# O-Doping Configurations Reduce the Adsorption Energy Barrier of K-Ions to Improve the Electrochemical Performance of Biomass-Derived Carbon

**DOI:** 10.3390/mi13050806

**Published:** 2022-05-22

**Authors:** Kai Zhao, Changdong Chen, Ming La, Chenghao Yang

**Affiliations:** 1College of Information Engineering, Pingdingshan University, Pingdingshan 467000, China; zhaokai@pdsu.edu.cn; 2School of Enviornment and Energy, South China University of Technology, Guangzhou 510006, China; lmccd5613@163.com; 3College of Chemistry and Environmental Engineering, Pingdingshan University, Pingdingshan 467000, China

**Keywords:** O-doping, biomass-derived carbon, K-ion batteries, energy storage mechanism

## Abstract

In recent years, atomic-doping has been proven to significantly improve the electrochemical performance of biomass-derived carbon materials, which is a promising modification strategy. Among them, there are relatively few reports about O-doping. Here, porous carbon derived from orange peel was prepared by simple carbonization and airflow-annealing processes. Under the coordination of microstructure and surface groups, the derived carbon had excellent electrochemical performance for the K-ion batteries’ anode, including a high reversible specific capacity of 320.8 mAh/g, high rate performance of 134.6 mAh/g at a current density of 2000 mA/g, and a retention rate of 79.5% even after 2000 long-term cycles, which shows great application potential. The K-ion storage mechanisms in different voltage ranges were discussed by using various characterization techniques, that is, the surface adsorbed of K-ionswas in the high-potential slope area, and the intercalation behavior corresponded to the low-potential quasi-plateau area. In addition, the density functional theory calculations further confirmed that O-doping can reduce the adsorption energy barrier of K-ions, change the charge density distribution, and promote the K-ion storage. In particular, the surface Faraday reaction between the C=O group and K-ions plays an important role in improving the electrochemical properties.

## 1. Introduction

With the large-scale application of Li-ion batteries (LIBs), the consumption of the limited Li resources in the continental crust is gradually increasing, the industry concentration is also high, and the resource monopoly pattern is obvious, which makes the corresponding raw material prices continue to rise [1,2]. In the process of looking for alternatives, abundant potassium salt has attracted increasingly more attention, which further drives the rapid development of K-ion batteries (KIBs). Compared with LIBs, KIBs also belong to the “rocking chair battery”, which has similar components, systems, and electrochemical reaction mechanisms, and is expected to become a large-scale energy storage device with excellent comprehensive efficiency [3].

Currently, carbon-based materials with unique properties have occupied an important position in the field of energy storage and environmental science [4,5]. However, due to the large radius of K-ions, the requirements for the structural stability and ion transport properties of anode materials are more stringent butthe traditional graphite cannot meet the requirements. Biomass-derived carbon is a kind of amorphous carbon with low cost and environmental protection characteristics and has been widely studied in LIBs [6]. Recent reports have also confirmed that biomass-derived carbon can be used as an anode for KIBs, such as derived carbon from chicken bone [7], bagasse [8], and corn silk [9], all of which show highly competitive electrochemical performances. However, compared with metal oxides and alloys, the capacity of biomass-derived carbon is relatively low and requires further improvement. Therefore, researchers have developed a series of modification strategies to solve this problem, in which atomic-doping is one of the simple and efficient modification methods.

Theoretically, there is a difference in electronegativity between the introduced heteroatoms and C atoms, which can improve the conductivity of biomass-derived carbon, provide more reactive sites, and enable it to have a low diffusion barrier, high charge distribution, and metal band gap [10]. These positive effects are beneficial to the adsorption and diffusion of K-ions, to improve the specific capacity, rate performance, and long-term stability. Taking an N atom as an example, its electronegativity is high. When occupying the sites of a carbon lattice, it will change the charge distribution and electronic properties of the carbon structure, induce the generation of additional defects, and promote the ion diffusion rate and electron conduction rate [11]. Wang et al. prepared hemp core-derived carbon with an N content of 8.56% using urea as an N source [12]. Compared with pure derived carbon, doping N increases the capacity from 115.2 mAh/g to 442.4 mAh/g, and the diffusion resistance decreases significantly. S atom doping can also improve the conductivity. Tian et al. reported an S-doped carbon-derived material from bamboo charcoal [13]. The introduced S mainly exists in the form of an -C-S-C- group, which can provide more adsorption sites. The derived carbon exhibits a rate capacity of 124.2 mAh/g, and the capacity can be maintained at 203.8 mAh/g after 300 cycles at a current density of 200 mA/g, showing excellent electrochemical performance. In addition, O-doping is also a common modification method, but in most cases, it is introduced in the form of self-doping and processes with O-rich precursors [14,15]. Xu et al. obtained O-doped-derived carbon by the simple direct heat treatment process using water chestnut as a carbon source [16]. The results showed that with the carbonization temperature increases, the O content of the products decreases gradually, resulting in a great difference in electrochemical performance. Yang et al. also analyzed the effect of O-doping on the K storage performance of the derived carbon anode by density functional theory (DFT) calculation [17]. Compared with the pure carbon, O-doped carbon shows a more negative binding energy, indicating that the adsorption capacity of K-ions is stronger, which can improve the storage capacity.

In fact, atomicdoping requires the use of chemical reagents, which may lead to environmental pollution problems. At the same time, biomass itself contains limited beneficial heteroatoms, which restricts the modification effect of derived carbon. Therefore, it is particularly important to develop simple and green doping technologies. Airflow-annealing is a common heat treatment technology in the field of metal processing and organic synthesis. Recently, sufficient experimental data have shown that airflow-annealing can introduce O-containing functional groups into materials [18]. In this work, O-doped orange peel-derived carbon (OPDC) with 13.38% O content was prepared by combining carbonization and airflow-annealing. Based on the comprehensive analysis of the electrochemical performance, the energy storage mechanisms of OPDC were discussed by using different characterization techniques. Moreover, DFT calculations were carried out for the effect of different O-doping configurations, which confirmed that the C=O group has an important significance to improve the electrochemical performance.

## 2. Results and Discussion

The preparation process of OPDC is shown in Figure 1a. It is well known that the morphology of derived carbon materials has a great influence on the electrochemical properties. As shown in Appendix A, OPDC is a three-dimensional porous skeleton structure, which is not only conducive to electrolyte penetration and provides more reactive sites, but can also alleviate the large volume change of materials during the charging/discharging process. The folded areas on the surface are connected with each other (Appendix A), which can also shorten the diffusion paths of electrons and ions, increasing the transport rate. Further analyzing the microstructure of OPDC, as shown in Appendix A, consistent with previous reports, the arrangement of the graphite-like microcrystalline structure presents an irregular disorder with only a few stacked regions [19]. Unlike the long-range ordered microcrystalline arrangement of graphite, this is mainly attributed to the fact that biomass precursors contain a large amount of oxygen, which will hinder the growth of microcrystalline graphite during carbonization [20]. Meanwhile, the carbon layer spacing of OPDC is 0.388 nm, which is larger than that of graphite. Considering the large radius of K-ions (0.133 nm), the expanded space between microcrystalline pseudo-graphite is theoretically more conducive to the intercalation and removal of K-ions, thus improving the K storage capacity. Furthermore, the fast Fourier transform (FFT) image shows no crystal diffraction spots, suggesting that the OPDC is a typical amorphous carbon material.

X-ray diffraction analysis (XRD) tests also confirmed the amorphous characteristics of OPDC, as shown in Figure 1b. Two diffraction peaks are found at ~24.3° and 44.1°, which correspond to the (002) and (100) crystal planes of the hexagonal graphite structure. The broadened diffraction peak shows that the crystallinity of OPDC is not ideal and belongs to typical nongraphitized carbon. The thickness of microcrystalline pseudo-graphite along the c-axis (L_c_, Appendix A) calculated by the Bragg formula is only 1.54 nm, indicating that only a small amount of stacked microcrystalline pseudo-graphite exist [21]. Moreover, by analyzing the information about molecular vibration and rotation, it can be seen that the D-band representing the degree of structural disorder is located at 1356 cm^−1^, while the G-band representing crystallinity and symmetry is located at 1593 cm^−1^ (Figure 1c), thus reflecting the amorphous state of OPDC. The composite strength ratio (I_G_/I_D_) of the G-band and D-band is 0.92, indicating that the material has a more disordered layer structure [22]. Moreover, based on the I_G_/I_D_, the length (L_a_, Appendix A) of microcrystalline pseudo-graphite along the a-axis is 17.66 nm, which can shorten the diffusion path of K-ions [23].

The N_2_ adsorption/desorption test can be used to obtain the specific surface area and pore structure of OPDC, which is particularly important for the electrochemical performance. As shown in Figure 1d, the obtained isothermal curve is IV-type and shows a certain hysteresis phenomenon in the relative pressure range of 0.4~1.0, showing that the pore structure of the material is irregular, and the mesoporous pore accounts for the largest proportion [24]. The specific surface area of OPDC calculated by the Brunauer/Emmett/Teller (BET) multi-point method is 113.86 m^2^/g. Compared with metal organic framework-derived carbon with high specific surface area, such a low-surface-area-derived carbon from biomass can limit the formation of SEI film, effectively reduce the consumption of active K-ions, and thus improve the initial Coulombic efficiency [25]. The pore size distribution of OPDC is shown in the illustration. It can be observed that there are micropores (0.79 nm), mesopores (2.75 nm), and macropores (68.50 nm) in the material, and the mesopores are the main ones. As is known to all, the micropores can be used to contain the charge in the electric double layer, providing more electrochemical reaction active sites for ion storage. The mesopores can offer a rapid ion diffusion path, enhancing the rate performance, while the macropores are beneficial to the penetration of the electrolyte and to the strengthening of the mechanical stability of the structure, which would play a positive role in the storage of K-ions.

In addition, the chemical bond or group state in the molecule of OPDC was analyzed by a Fourier transform infrared spectrometer (FTIR), as shown in Appendix A. The transmission peak at about 3400 cm^−1^ is formed by the vibration of O-H bonds of alcohol, phenol, and carboxylic acid, but it is lower than the normal value, which may be affected by hydrogen bonds. There are several characteristic peaks with small area in the wavenumber range of 3000 to 2800 cm^−1^, which correspond to the aliphatic C-H bond. The transmission peaks of the C=O bond, C=C bond, aromatic C-O bond, and aliphatic C-C bond are distributed in the wavenumber range of 1800 to 800 cm^−1^ [26,27]. After high-temperature heat treatment, the infrared rays are basically absorbed, indicating that the group content of OPDC is sharply reduced, which is mainly related to the formation of volatile gases by a large number of intramolecular and intermolecular O-H bond, C-H bond, and C-O bond fractures. At this time, the biomass has been completely transformed into derived carbon.

In order to understand the chemical composition and configuration of OPDC, anX-ray photoelectron spectrometer (XPS) test was carried out. As shown in Figure 1e, two characteristic peaks appear at 284.8 and 532.4 eV in the spectrum, representing C 1s and O 1s, respectively. After the low-temperature O-doping process, the O content of OPDC reaches 13.38%. In addition, no characteristic peaks of elements except for C and O are observed, indicating that the product after pickling has high purity. Figure 1f shows the high-resolution C 1s XPS spectra of OPDC, which mainly appears in three configurations. The C-C group in the defect-free graphite lattice is located at 284.8 eV, and the peaks at 285.7 and 288.9 eV are attributed to the C-O group in the defective graphite lattice and C=O group [28]. It is noteworthy that no characteristic peak indicating a π–π interaction is detected in the spectra, indicating that only a small amount of microcrystalline pseudo-graphite isstacked together [29]. Meanwhile, as shown in Figure 1g, O 1s spectra can be deconvoluted into two characteristic peaks at 531.9 and 533.4 eV, corresponding to the broadening signals associated with the presence of C atoms singly or doubly coordinated with an O atom (O=C group and O-C group) [30,31]. In the total O, the contents of the O=C group and O-C group are 57.80% and 42.20%, respectively. As mentioned above, the surface Faraday reaction between the C=O group and K-ions can be converted to C-O-K, similar to Na-ion [32]. This capacitive process is highly reversible, thus improving the cycling capacity, especially at large rate.

For the electrochemical performance tests, Figure 2a shows the cyclic voltammetry (CV) curves of the OPDC electrode at the scanning rate of 0.1 mV/s. In the first cycle, two reduction peaks appear at 0.76 and 0.39 V. The former is related to the side reaction between the electrolyte and the surface groups, while the latter is caused by the decomposition of the electrolyte and the formation of a solid electrolyte interface (SEI) film at the interface [33]. These two reduction peaks disappear in the subsequent cycle, indicating that the irreversible reaction mainly occurs in the first charge/discharge process. A pair of obvious redox peaks can be observed in the low-potential region close to 0 V, which is characterized by the intercalation and removal of K-ions in the carbon layer [34]. The vague broad oxidation peak near 0.86 V corresponds to the desorption behavior of K-ions on the microcrystalline pseudo-graphite surface, similar to the capacitance characteristics [35]. At the same time, the CV curves almost overlap in the subsequent cycle, which means that the intercalation/removal and adsorption/desorption behaviors of K-ions have good reversibility.

The typical galvanostatic charge/discharge curve is shown in Figure 2b, and there is no obvious plateau area. At the current density of 50 mA/g, the specific capacities of initial discharge and charge of the OPDC electrode are 508.1 and 320.8 mAh/g, respectively, and the corresponding Coulombic efficiency is 63.1%. A large amount of capacity loss is mainly related to irreversible SEI film formation. After 150 cycles, the reversible capacity of the OPDC electrode can be maintained at 259.5 mAh/g (Figure 2c), showing excellent cycling performance. On the other hand, an electrochemical impedance spectroscopy (EIS) test was used to further confirm the excellent cycling ability. Figure 2d shows the Nyquist plots of the OPDC electrode at different cycles and provides an equivalent fitting circuit model. All plots consist of a point in the high-frequency region, a semicircle in the high-middle-frequency region, and an inclined straight line in the low-frequency region. It is well known that the intercept with the *x*-axis represents the ohmic resistance (R_o_), mainly including the inherent resistance of OPDC and electrolyte, as well as the contact resistance at the interface between the collector and the active material. The semicircle corresponds to the resistance of K-ions across the SEI film (R_SEI_) and the charge transfer resistance at the OPDC/electrolyte interface (R_ct_). The inclined straight line belongs to Warburg impedance (Z_w_), which is related to the solid diffusion process of K-ions inside the material [36]. CPE represents the capacitance for both the double layer and passivation film [37]. The kinetic parameters obtained by fitting are summarized in Appendix A. It can be observed that R_SEI_ changes little during the cycle, which can be interpreted as the result of stable SEI film formation. However, the R_ct_ decreases from the first cycle (1406.00 Ω) to the 10th cycle (685.21 Ω), which seems to be understood as the activation factor of the electrode. With the increase in cycles, the R_ct_ continues to increase, suggesting that the intercalation/removal activity of K-ions would gradually decrease, which is consistent with the results previously reported. It can be seen that the reduction in specific capacity during the cycle is mainly attributed to the consumption of active ions and the reduction in reaction kinetics caused by the formation of the SEI film.

Compared with common carbon-based materials, it shows a poor rate performance in the K-ion intercalation/removal process. The OPDC electrode has a good capacity response to the current density change from 50 to 2000 mA/g, as shown in Figure 2e. Even at a high rate of 2000 mA/g, the reversible capacity of 134.6 mAh/g can still be maintained. In particular, when the current density is restored to 50 mA/g, most of the specific capacity is recovered, which proves that the OPDC electrode also has a stable cycle process at high rate. As shown in Figure 2f, the long-term cycle stability of the OPDC electrode was further tested. At 2000 mA/g, there is no obvious attenuation for more than 2000 cycles, the capacity retention rate is 79.5%, and the Coulombic efficiency is almost 100%. Then, the electrochemical performances of OPDC and some biomass-derived carbon were compared, as shown in Appendix A. It can be observed that each index of the OPDC electrode is excellent, the low specific surface area can reduce the consumption of active K-ions and improve the initial Coulombic efficiency, surface O-doped groups can increase defect sites and promote the surface adsorption contribution of K-ions, and the larger carbon layer spacing is more conducive to the smooth intercalation and removal of K-ions, improvingthe ratio performance [38].

The reaction kinetics of the OPDC electrode was analyzed by various sweep rate CV teststo further explore the K storage mechanisms. As shown in Figure 3a, when the scanning rate increases from 0.3 to 1.0 mV/s, the peak current representing the intercalation and removal of K-ions gradually increase, and the corresponding potential also shifts, indicating that the reaction process of the electrode is close to the characteristics of surface adsorption [39]. It is assumed that the peak current (*i*) is exponentially related to the scanning rate (*v*) [40], i.e.,
(1)$$i=av^{b}$$
where a and b represent constants. In particular, when the b value is close to 1, the electrochemical reaction of the electrode is dominated by capacitive behavior, such as the reaction between the K-ion and the C=O group. When the b value is equal to 0.5, the capacity contribution is controlled by the diffusion process [41]. Figure 3b shows the relationship between the peak current of the OPDC electrode and the scanning rate. The slope of the fitted curve, i.e., the b value, is equal to 0.69, which means that the K storage mechanism of OPDC is a mixed mechanism containing capacitive behavior and diffusion process control.

In addition, according to the previous reports, the capacity contribution rate of capacitive behavior can also be quantitatively analyzed by using the following formula [42]:(2)$$i=k_{1}v+k_{2}v^{\frac{1}{2}}$$

In the current response, k_1_*v* corresponds to the capacitive behavior, and k_2_*v*^1/2^ is the diffusion control characteristic. The slope (k_1_) and intercept (k_2_) can be obtained by drawing the corresponding curve. The CV curve and capacitive behavior curve at the scanning rate of 0.7 mV/s are shown in Appendix A, and the calculated contribution rate is 52.9%. Figure 3c depicts the relationship between the scanning rate and capacitive contribution rate. It can be found that when the scanning rate is 0.3 mV/s, the capacitive contribution ratio of the OPDC electrode is 37.0%, while when the scanning rate increases to 1 mV/s, the capacitive process accounts for 63.7%. These results indicate that the high contribution ratio induced by the K-ion adsorption behavior of the OPDC electrode is mainly induced by the active sites provided by the porous structure and O-doping, which plays a positive role in improving the K-ion adsorption capacity and promoting mass transfer kinetics, thus explaining the reason for the good ratio performance.

Here, the surface adsorption and carbon layer intercalation processes of K-ionsare collectively referred to as diffusion in a broad sense, which is an important form of mass transfer. The apparent diffusion coefficient of K-ions (D_K_) directly affects the reaction rate and the overall performance. Therefore, the D_K_ value of OPDC was calculated by the galvanostatic intermittent titration technique (GITT) test, which is of great significance for the analysis of electrochemical kinetics. Figure 3d shows the GITT curve of the OPDC electrode after the third charge/discharge cycle. D_K_ can be estimated by the following simplified formula [43]:(3)$$D_{K}=\frac{4}{\pi \tau }{\left(\frac{m_{B}V_{M}}{M_{B}S}\right)}^{2}{\left(\frac{\Delta E_{s}}{\Delta E_{\tau }}\right)}^{2}$$

Among them, *τ* is the pulse duration, *m_B_* and *V_M_* represent molar volume and molecular mass, respectively, *M_B_* corresponds to the mass of OPDC, and *S* is the effective contact area between electrode and electrolyte. The thickness of OPDC was tested by SEM, and the rough substitute formula is (*m_B_V_M_*/*M_B_S*). ∆*E_s_* is the potential difference of quasi-thermodynamic equilibrium before and after the pulse, and ∆*E**_τ_* is the potential difference during the pulse (Figure 3e). According to the voltage-diffusion coefficient curve (Figure 3f), the D_K_ of OPDC is of the order of 10^−10^ cm^2^/s.

The theoretical calculation results show that the diffusion energy barrier of K-ions between the surface-active sites is almost 0, which explains why D_K_ always maintains a high value in the high-potential slope area. As the K-ions adsorbed on the active site reach saturation, the remaining K-ions begin to intercalate in the carbon layer. This process requires the energy barrier to be overcome between microcrystalline pseudo-graphite and the repulsive force of pre-adsorbed K-ions, resulting in an increased diffusion difficulty, which is manifested in the sharp decline in D_K_, and corresponds to the change characteristics of low-potential quasi-plateau area. The change trend of D_K_ during the charging/discharging process is almost completely opposite, indicating that the whole electrochemical reaction process is highly reversible. The capacity increase caused by O-doping is mainly reflected in the high-potential slope area.

The results show that the capacitive behavior and the carbon layer diffusion characteristics also have different effects on the microstructure of the material. Theoretically, the adsorption induction process of K-ions mainly reacts with the active sites (e.g., C=O group) without changing the carbon structure. The diffusion intercalation process of K-ions combines with C atoms to form compounds, which causes the distance between carbon layers to expand. As shown in Appendix A, the electrodes with different cut-off voltages were selected for the XRD test in the charging/discharging process to further distinguish the voltage range corresponding to the above electrochemical reaction mechanism. In the high-potential slope area, the (002) crystal plane diffraction peaks of a and b do not change, indicating that the carbon layer spacing is unchanged, corresponding to the capacitive behavior. The diffraction peak of the crystal plane gradually shifts to a small angle when the discharge continues to c and d in the low-potential quasi-plateau area, indicating that the carbon layer spacing is expanded, which belongs to the diffusion intercalation characteristics of K-ions. During the charging process, the (002) crystal plane diffraction peaks of sites e, f, g, and h show a reverse trend, and return to the initial state, which verifies the reversibility of the electrochemical reaction. These results are consistent with those reported by Liu et al. [44]. Moreover, Raman spectroscopy can also be used to characterize the storage process of K-ions. As shown in Appendix A, in the initial discharge stage, the positions of the G-band and D-band do not change. However, in the low-potential quasi-plateau area, the G-band and D-band gradually red-shift, and the peak intensities weaken, which is mainly related to the increase in electron density caused by K-ion insertion into the carbon layer [17].

For the capacitive contribution, O-containing groups are of great significance to improve the electrochemical performance. Therefore, applying first-principles theory, the effect of O-doping configuration on K-ion storage was studied by DFT calculation. The calculation methods in the Appendix A provide details. As shown in Figure 4a–c, the E_ads_ of K-ions on pure carbon, C-O, and C=O are calculated to be −0.19, −1.18, and −1.33 eV, indicating that both C-O and C=O groups can increase the adsorption capacity of the carbon material for K-ions. The calculated total energy of the species is summarized in Table 1. In contrast to the deformation charge densities of pure carbon, C-O, and C=O (Figure 5a–c), there is an electron-rich state near the O atom, which is mainly related to the negative polarization of the O atom caused by the electronegative difference between the C atom and O atom, thus significantly improving the adsorption capacity of K-ions in the electrochemical reaction process [16]. Meanwhile, the density of states (DOS) of pure carbon, C-O, and C=O are shown in Figure 6. Compared with pure carbon, both C-O and C=O group-doped carbon have more electron enrichment at the Fermi level, and a higher DOS, which will improve the conductivity of the material. Among them, the change of C=O group-doped carbon is the most obvious. Therefore, O-doping has an indispensable influence on the K storage capacity of OPDC. However, according to the previous reports, the increase in C-O group content will lead to more irreversible capacitance, which is also one of the factors that reduce the initial Coulombic efficiency [45]. Therefore, future research can focus on increasing the C=O group content, such as adding chemical dopants.

Finally, Appendix A depicts the K storage mechanisms of OPDC. In the high-potential slope area, the surface Faraday reaction between the C=O group and K-ions plays an important role in K storage, as well as the adsorption of K-ions at the defects. The surface induction characteristics mainly provide high capacitance and fast kinetics. In the low-potential quasi-plateau area, the capacity is contributed by K-ions intercalated in the microcrystalline pseudo-graphite layer.

## 3. Conclusions

In summary, this work designs an OPDC modified by airflow-annealing as an O-doping process. Surface O-rich groups (13.38%) can increase the reactive active sites and significantly improve the capacitive capacity. Under the synergistic effect, the reversible capacity of OPDC at 50 mA/g is 320.8 mAh/g, and the high rate performance is 134.6 mAh/g. After 2000 cycles, the capacity retention rate can still be 79.5%, which has excellent electrochemical performance. At the same time, various scanning rates, CV, GITT, and XRD also confirm that the capacity contribution of the high-potential slope area is from the surface adsorption behavior of K-ions, while the low-potential quasi-plateau area represents the characteristics of K-ions intercalating between carbon layers. It is noteworthy that the theoretical calculation results show that O-doping can improve the adsorption energy of K-ions and change the conductivity, which is beneficial to the K storage. Among them, the C=O group has the best effect.

## Figures and Tables

**Figure 1 micromachines-13-00806-f001:**
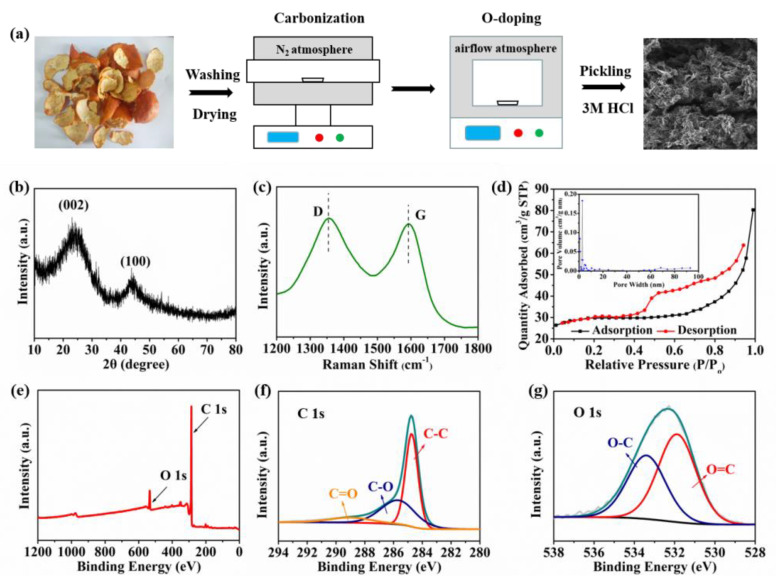
(**a**) Preparation scheme of OPDC. (**b**) XRD pattern. (**c**) Raman spectrum. (**d**) N_2_ adsorption/desorption isotherms and pore size distribution. (**e**) Survey XPS spectrum, (**f**) high-resolution C 1s XPS spectra, and (**g**) high-resolution O 1s XPS spectra of OPDC.

**Figure 2 micromachines-13-00806-f002:**
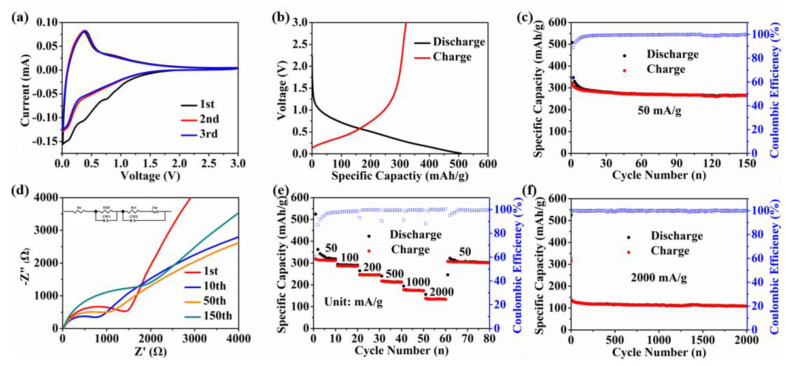
(**a**) CV curves scanned at a rate of 0.1 mV/s. (**b**) Galvanostatic charge/discharge curve at a current density of 50 mA/g. (**c**) Cycling performance. (**d**) Rate capability. (**e**) Long−term cycling stability. (**f**) Nyquist plots and equivalent circuit.

**Figure 3 micromachines-13-00806-f003:**
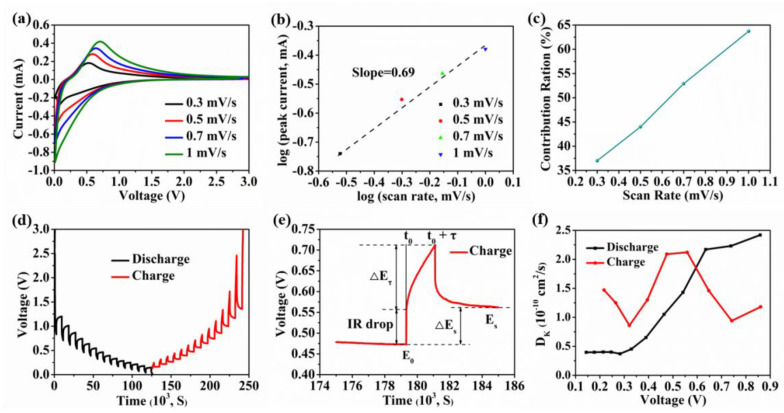
(**a**) CV curves at various scan rates. (**b**) Relationship between the peak current and scanning rates. (**c**) Capacitance contribution at different scan rates. (**d**) GITT curve. (**e**) A single GITT titration curve. (**f**) K−ion apparent diffusion coefficients for the discharge and charge processes.

**Figure 4 micromachines-13-00806-f004:**
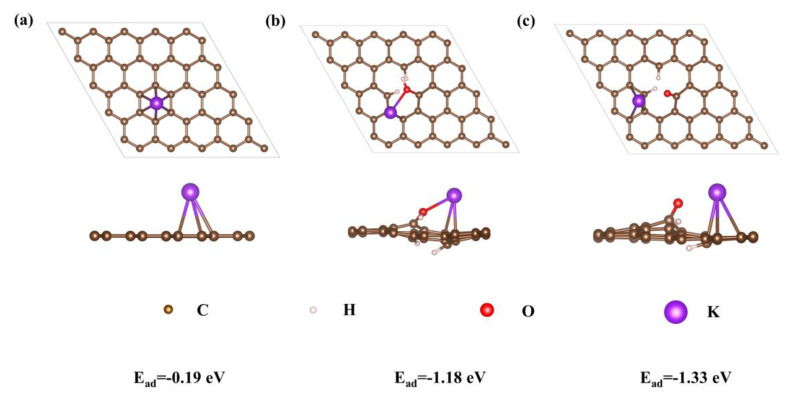
Adsorption of K−ions on (**a**) pure carbon, (**b**) C−O, and (**c**) C=O and the corresponding side views. Brown indicates C atoms, pink indicates H atoms, red indicates O atoms, and purple indicates K atoms.

**Figure 5 micromachines-13-00806-f005:**
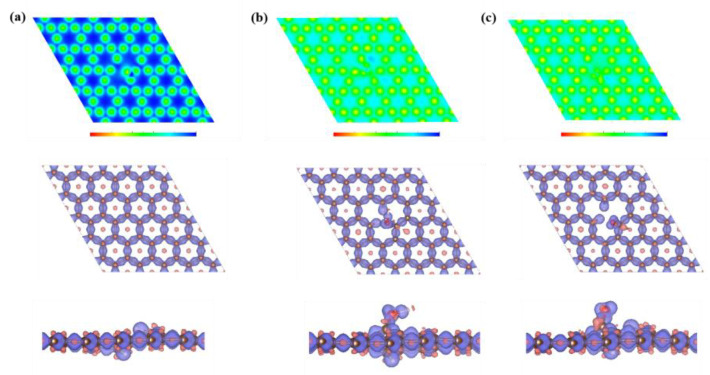
Deformation charge density maps of (**a**) pure carbon, (**b**) C-O, and (**c**) C=O and the corresponding side views.

**Figure 6 micromachines-13-00806-f006:**
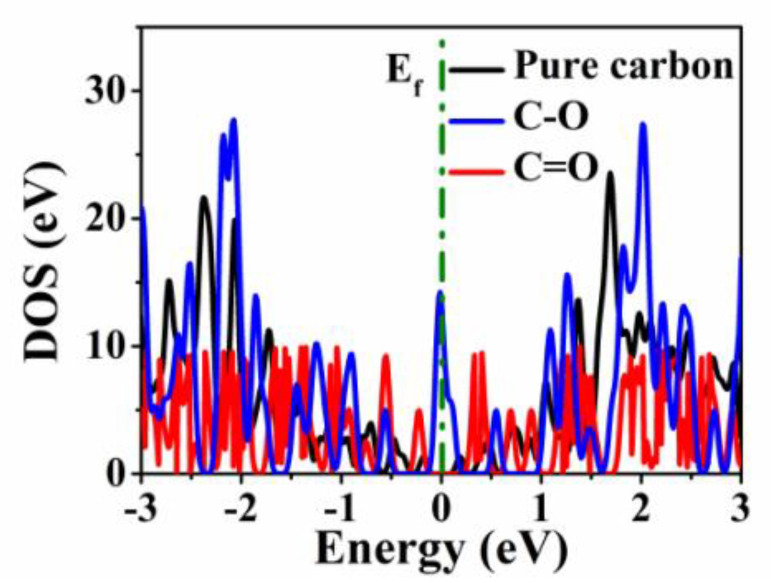
DOS for pure carbon, C=O-doped carbon, and C−O−doped carbon.

**Table 1 micromachines-13-00806-t001:** Total energy of the species calculated.

	E_surf_ (eV)	E_total_ (eV)	E_K_ (eV)	E_ads_ (eV)
Pure carbon	−461.53	−462.77	−1.05	−0.19
C=O	−462.48	−464.86	−1.05	−1.33
C-O	−465.88	−468.11	−1.05	−1.18

## Data Availability

The data presented in this study are available on request from the corresponding author.

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
