# Peer review of "O-Doping Configurations Reduce the Adsorption Energy Barrier of K-Ions to Improve the Electrochemical Performance of Biomass-Derived Carbon"

_micromachines, 2022, doi:10.3390/mi13050806_

Round 1
Reviewer 1 Report
This work presented a biomass-derived O-doped carbon materials which showed good performance as anode materials for K-ion batteries. DFT was used to confirmed the effects of O doping on K-ion storage. Overall, this is an interesting work. However, some supplementary experiments are suggested before it can be accepted.
- How did the carbon structure change after stored K ions? XRD pattern and Raman spectrum of the carbon anode in potassiation state are suggested to be supplied.
- The author claimed that O-doping played an important role in K-ion storage. What is the performance of the carbon anode if the O heteroatoms were removed by heat-treatment in inert gas?
- Some important works about heteroatom doping for K-ion storage mechanism are helpful to analyze the O doping for the K-ion storage of the carbon anode: e.g., J. Mater. Chem. A, 2022, 10, 9612-9620; Carbon, 2021, 178, 775-782
Reviewer 2 Report
Zhao et al have reported the O-doping biomass-derived carbon to improve the electrochemical performance as KIBs. By carbonization and airflow-annealing process that is easy and green, they successfully achieved O-doping porous carbon, presenting excellent electrochemical performance. The K-ion storage mechanism is revealed by different electrochemical characterization techniques and DFT calculations, suggesting that O-doping can reduce the adsorption energy barrier of the K-ion and improve the conductivity. In addition, they find the surface Faraday reaction between C=O group and K-ions plays an important role in improving the electrochemical properties.
The work presented in this paper is of good quality, covering both experimental materials science and calculations. The writing is clear and precise. I think the results will attract wide readers of the journal, and therefore would like to recommend publication if the authors answer the following comments successfully.
[1] O-content in OPDC is 13.38%. Has the author controlled the content of the oxygen to optimize the electrochemical performance?
[2] To present the good performance of the OPDC, it is suggested to compare the electrochemistry performance with other biomass derived carbon with different doping via table or graph.
[3]Please indicate the type of atoms with different color in Fig. 4.
[4] In Fig. S1c, the scale bar is wrong and the inset is more-likely a FFT pattern other than the SAED.
[5] The discussion and the corresponding caption of Fig. 4-6 are too simple. Please enrich them.
[6] The C-O and C=O has different influence on the performance. Please give some discussions on how to control the content of them so that guide the further material synthesis and optimization of the performance.
Reviewer 3 Report
This manuscript reports O-doped orange peel derived carbon (OPDC) with 13.38% O.
The authors stated that the introduction of O can boost the capacity. The reviewer recommends it to be published in this journal after some revisions.
- The authors state that the introduction of O can improve the capacity of orange peel derived carbon. Though theoretical calculation is used to demonstrate this opinion, the experimental results is not enough. The orange peel derived carbon without O-doping should be made to compare the difference. It is more direct to affirm the influence of O-doping.
- The process need two steps of annealing. And the statement of “simple and green doping technologies” may be not accurate.
- The description for FTIR results (Figure S3) is not suitable. The transmission peaks for the bonds should be clear.
- Many format problems should be modified. Such as “N2” “Based on the IG/ID” “113.86 m2/g” “3400 cm-1” “,MB S …” should be italic. The authors should check the whole manuscript.
- Some expressions are not fluent. For example “Further analyze the microstructure of OPDC, as shown in Fig. S1c”.
Round 2
Reviewer 3 Report
The authors have addressed my comments, therefore, I would recommend its publication in present form.